# Precisely spun super rotors

Ivan O. Antonov [1], Patrick R. Stollenwerk [1], Sruthi Venkataramanababu[2], Ana P. de Lima Batista [3], Antonio G. S. de Oliveira-Filho [3] & Brian C. Odom [1,2 ✉]

Improved optical control of molecular quantum states promises new applications including chemistry in the quantum regime, precision tests of fundamental physics, and quantum information processing. While much work has sought to prepare ground state molecules, excited states are also of interest. Here, we demonstrate a broadband optical approach to pump trapped $SiO^+$ molecules into pure super rotor ensembles maintained for many minutes. Super rotor ensembles pumped up to rotational state $N = 67$, corresponding to the peak of a 9400 K distribution, had a narrow $N$ spread comparable to that of a few-kelvin sample, and were used for spectroscopy of the previously unobserved $C^2\Pi$ state. Significant centrifugal distortion of super rotors pumped up to $N = 230$ allowed probing electronic structure of $SiO^+$ stretched far from its equilibrium bond length.

[1] Department of Physics and Astronomy, Northwestern University, Evanston, IL, USA. [2] Applied Physics program, Northwestern University, Evanston, IL, USA. [3] Departamento de Química, Laboratório Computacional de Espectroscopia e Cinética, Faculdade de Filosofia, Ciências e Letras de Ribeirão Preto, Universidade de São Paulo, Ribeirão Preto-SP, Brazil. ✉email: b-odom@northwestern.edu

Super rotors are molecules with rotational energy that greatly exceeds $kT$ and may approach or exceed the bond energy[1,2]. Super rotors can be used for isotope separation[1], optical deflection[3], probing of molecular structure far from equilibrium geometry[4], and controlled dissociation of molecular bonds[5]. They are known to possess unique collisional relaxation pathways[6], anisotropic transport properties[7], and surface scattering patterns[8], and are expected to form macroscopic vortex flows upon relaxation[9]. Super rotors were detected in interstellar clouds where they form upon photodissociation of molecules by high-energy photons[10].

Super rotors are challenging to produce in the laboratory. Sufficiently hot thermal samples contain very broad state distributions, and would often create environments in which the molecules are unstable. Super rotors have been produced by chirped high-intensity fields in optical centrifuges, where the dynamics are understood as response to a classical "corkscrew" potential or alternately as from a sequence of Raman transitions[2]. While this is a very elegant approach which works for a broad class of molecules, the Liouville theorem implies that without dissipation, entropy cannot be reduced. Consequently, the rotational distribution is not narrowed, and if it begins hot then a significant fraction of the molecules are lost from the centrifuge as it spins up[1,2]. In addition, it is sometimes observed that the strong non-resonant optical fields cause unwanted excitations or photochemistry even for closed-shell diatomics[6], phenomena that will be widespread in more complex systems.

Optical pumping represents a different approach for state control of quantum systems. Following resonant excitation by a low-intensity laser, spontaneous emission can sink entropy and narrow the state distribution. Although lowering entropy can sometimes be achieved in stimulated Raman processes[11], using spontaneous emission is often far simpler and serves as a workhorse technique for cooling atoms. Recently, optical pumping has also been used to cool molecules to their ground vibrational[12] and rotational[13–15] manifolds, and toward particular hyperfine states[16]. It has previously been proposed to use many lasers, generated by Raman processes in a gain medium, for pumping to super rotor states[17].

Here, we demonstrate the first preparation of trapped super rotors by means of optical pumping. Taking into consideration angular momentum selection rules, it is not at first obvious that optical pumping can be used to create super rotor states. However, this goal can be achieved by driving many repeated cycles of absorption and spontaneous emission on subsequent spectral lines. To this end, we use a broadband spectrum, which is filtered[15] such that no absorption occurs from the target rotational state $N$, but other states have their P or R branch transitions illuminated such that population is driven toward the target state. $SiO^+$ was previously proposed[18] as a favorable candidate for optical pumping using the $B^2\Sigma^+$-$X^2\Sigma^+$ electronic transition. This transition has two favorable properties: a short spontaneous emission lifetime of 74 ns[19] for fast pumping, and highly diagonal Frank–Condon factors (FCFs) for decoupling vibrational and electronic excitations, thereby simplifying state control[20]. The $SiO^+$ are confined to a linear Paul trap. They are initially rotationally hot, and pumping narrows the entire distribution to a few rotational levels about the target. We use these narrow ensembles of super rotors to probe molecular structure far from the equilibrium bond length and to perform spectroscopy of a previously unobserved excited electronic state.

## Results

**$SiO^+$ trapping and state analysis.** $SiO^+$ ions were stored in a linear Paul trap mounted inside an ultrahigh vacuum chamber (Fig. 1b). In order to increase trapping times and to reduce the volume over which optical pumping was required, $SiO^+$ samples were sympathetically cooled into Coulomb crystals by co-trapped laser-cooled $Ba^+$ ions. A typical Coulomb crystal in the experiment was comprised of 500–1000 $Ba^+$ ions with the inner core of 10–100 $SiO^+$ ions. The crystallized $SiO^+$ have translational temperatures as low as a few millikelvins, but their rotational temperature is not affected by $Ba^+$ sympathetic cooling. Without sympathetic cooling, $SiO^+$ had trap lifetimes of 10–30 s, while in the Coulomb crystal lifetimes were limited by reactions with background hydrogen, typically 10–20 min.

Reaction with background hydrogen irreversibly removes $SiO^+$ population via formation of $SiOH^+$[21]. Other external interactions that affected the rotation distribution of the super rotors (inelastic collisions with background gas, spontaneous emission decay to lower rotational levels, and interaction with blackbody radiation) were several orders of magnitude slower than the optical pumping rate and could easily be overcome to dynamically maintain the target state.

To characterize $SiO^+$ rotational control, we performed action spectroscopy using dissociation through the previously unobserved double-well $2^2\Pi$ state[19], henceforth referred to as $C^2\Pi$. Rotationally resolved excitation of quasi-bound levels in the inner well results in predissociation, whereby trapped $SiO^+$ converts to trapped $Si^+$. The wavelength dependence of predissociation efficiency, monitored using laser cooled fluorescence mass spectrometry (LCFMS)[22] produced a dissociation spectrum, which in turn revealed the rotational distribution of the original $SiO^+$ ensemble.

**Super rotor spectroscopy at high $N$.** The rotational lines of the $SiO^+$ $B^2\Sigma^+$-$X^2\Sigma^+$ transition (Fig. 1d) are well separated into P and R branches that obey $\Delta N = +1$ or $-1$ selection rules, respectively, that is increasing or decreasing rotation. Broadband pumping of only the P-branch can be used to cool $SiO^+$ rotations to $N = 0$[20]. More general tailored spectra driving transitions down to a selected $N$ using the P branch and up to the same $N$ using the R branch were used to pump population to excited rotational states. Broadband light near 385 nm produced by frequency doubling of an 80 MHz pulsed femtosecond laser was dispersed on a grating and focused onto a mask to remove unwanted frequencies (Fig. 1c). The spectrally filtered light was then back-reflected from a mirror, recombined on the same grating and sent into the ion trap.

To determine the spin-orbit and vibrational structure of $C^2\Pi$, and to make a first observation of the effects of optical pumping, we took a survey spectrum of the $C^2\Pi$ - $X^2\Sigma^+$ ($v'$, $v = 0$) bands, that is driving to various vibrational levels of $C^2\Pi$ (Fig. 2a). Broad bands were recorded with internally hot $SiO^+$ immediately after loading while a narrowing was observed after pumping toward $N = 0$, due to successful narrowing of the rotational distribution. A finer sweep of the $C^2\Pi_{1/2}$ - $X^2\Sigma^+$, 0–0 band (Fig. 2c) revealed rotationally hot spectra without pumping and clean resolved spectra after pumping toward $N = 0$, 10, 25, 40, 55, and 67. The narrow rotational distributions result in very simple spectra with well separated rotational branches, which were analyzed to determine rotational populations (Fig. 2d, e, Methods).

As in ref. [23], these narrow distributions were crucial for understanding $C^2\Pi$-$X^2\Sigma^+$ spectra at high $N$. Fig. 3 compares a spectrum of $SiO^+$ pumped toward $N = 67$ with a thermal spectrum at $T = 4600$ K, chosen to maximize population in that state. The rotational lines are ~1 cm$^{-1}$ broad due to fast predissociation, but they are resolved to the baseline and easily identified. In contrast, the simulated spectrum is an unresolved envelope of many lines originating from significantly different $N$. The number of populated rotational energy levels in

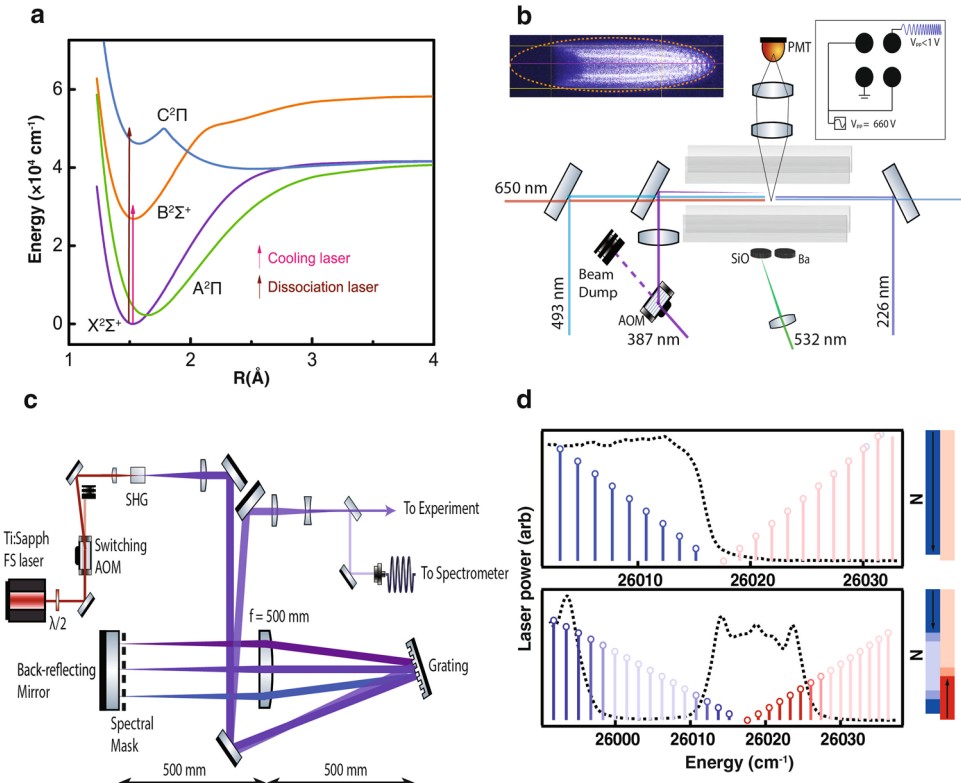

**Fig. 1 Experimental Overview. a** Relevant electronic states of $SiO^+$. **b** Experimental setup, a bright barium ion Coulomb crystal with a dark $SiO^+$ core is loaded into a linear Paul trap by 532 nm ablation, and translationally cooled using 493 nm and 650 nm $Ba^+$ transitions. $SiO^+$ is pumped with 385 nm and probed with 226 nm. A low-amplitude chirped RF waveform is used for LCFMS detection. **c** Spectral filtering setup for the 385 nm broadband light. **d** Spectrum of the 385 nm light used for pumping $SiO^+$ toward $N = 0$ (top) and $N > 0$ (bottom). Blue (red) sticks represent transitions in the P (R) branch. Arrows to the right show the flow of optical pumping, with blue (red) representing lowering (raising) $N$, and shading represents intensity of the light.

the optically pumped sample is comparable to a thermal population at $T = 3.6$ K.

As a first application of pure ensembles of super rotors, we extracted parameters for the $C^2\Pi$ structure. C-X spectra with $SiO^+$ pumped toward various $N$ were recorded for $v = 0$, 1, and 2 of both $C^2\Pi_{1/2}$ and $C^2\Pi_{3/2}$ (see Supplementary Table S2 for the line list). The spectra were fitted to the energy expressions for a $^2\Pi$-$^2\Sigma^+$ transition, where the $X^2\Sigma^+$ parameters were fixed at the literature values[24] and the $C^2\Pi$ parameters (Table 1) were determined from the fit. It is interesting to note that, although measurement of the centrifugal distortion at thermal $N$ would require high-resolution instrumentation, its extreme $N^4$ scaling makes it easy to measure in super rotors. The value of the centrifugal distortion of the C state is slightly higher than those of X and B states, which is consistent with its longer equilibrium bond length. Its agreement with Kratzer relationship[25] signifies that the C state potential near equilibrium is well described by a Morse potential.

**Dissociation at very high N.** Pumping to higher $N$ requires a modified protocol. The B-X transition R-branch bandhead near $N = 80$ puts conflicting requirements on a spectral filtering mask, which prevented preparing super rotors between $N = 80$ and $N = 160$ (Fig. 4). Super rotors with $N > 160$ were prepared using a two-step dynamic mask. In Step 1, the R-branch portion between the band origin and the $N = 160$ line was pumped. At this step, we completely avoided pumping the P-branch and waited several seconds to ensure that all $SiO^+$ molecules reached the $N = 160$ target rotational state. In Step 2, an additional portion of the R branch was exposed with a new cut-off positioned between R

(170) to R(230) (one of the pink areas in Fig. 4). These spectra also cover low-lying P-branch lines; however, since the population was pumped to high $N$ in the first step, the P-branch light has no effect. We did not attempt to perform $C^2\Pi$ spectroscopy for $N > 67$ because calculations indicated that its lifetime at these $N$ is too short to resolve rotational structure.

As a second application of pure super rotor ensembles, at these higher $N$ where centrifugal distortion causes a significant stretching of the bond length, we observe the onset of dissociation at $N > 190$. This behavior serves as a probe molecular structure far from the equilibrium geometry (Figs. 5, S2, S3). The dissociation rate is a function of $N$ because of the centrifugal force; this can be represented in the potential energy curve plots by the addition of a centrifugal potential term which distorts the curves.

During optical pumping, molecules occupy both the $X^2\Sigma^+$ and $B^2\Sigma^+$ states. Although for this range of $N$ the molecules are bound in the $X^2\Sigma^+$ state, dissociation from $B^2\Sigma^+$ can occur through coupling to the ground state asymptote via interactions with excited electronic states, such as $1^4\Sigma^-$ or $1^4\Pi$ (Fig. 5). In $SiO^+$, such beyond Born-Oppenheimer interaction occurs only when the bond is significantly stretched—normally requiring excitation to vibrational state $v \sim 10$. However, the centrifugally distorted super rotors interact noticeably even in $v = 0$. See Supplementary Information Section 2 for further discussion.

## Discussion

The super rotor rotational state distribution widths achieved here were similar to those of supersonic expansion or cryogenic buffer gas cooling. However, the optically pumped distributions can be centered at very high rotational energies. Ensembles of trapped

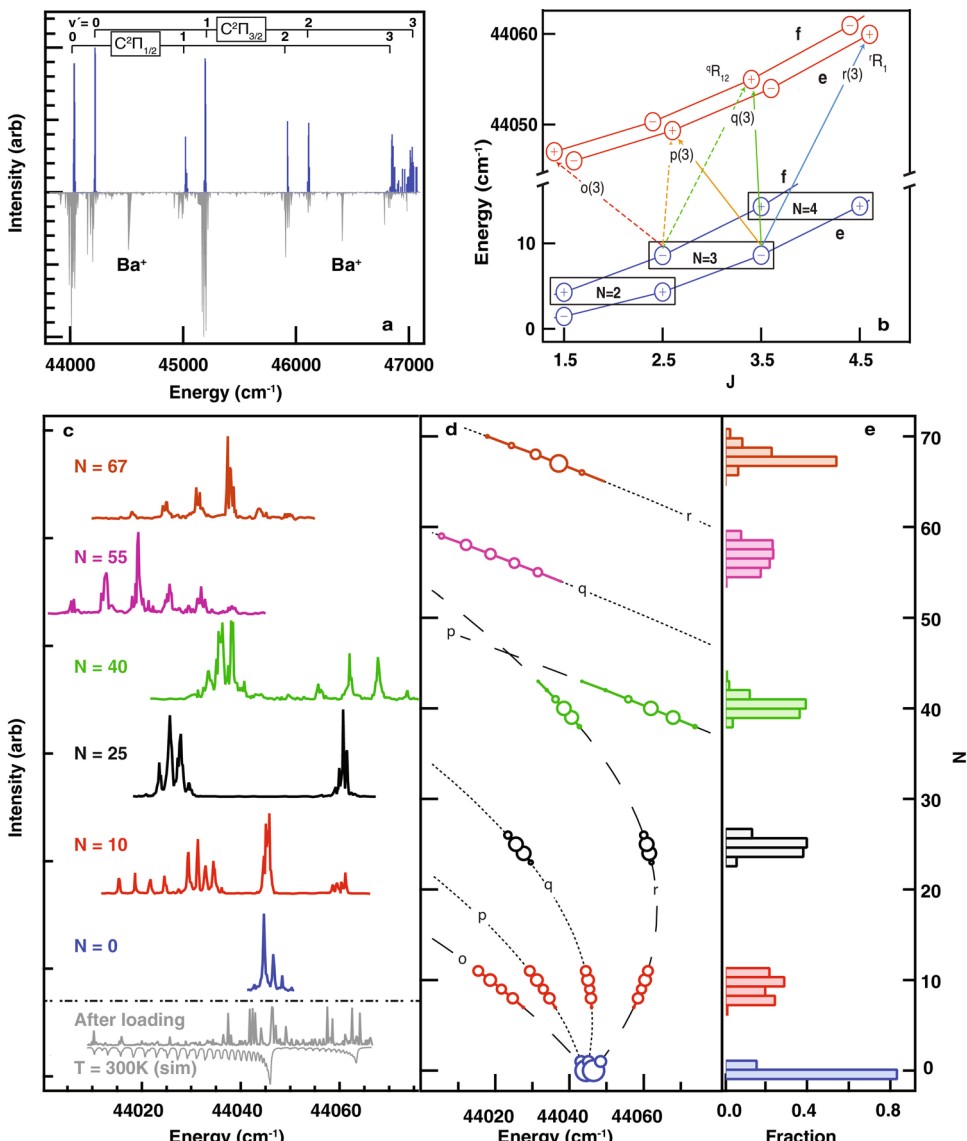

**Fig. 2 Super rotor spectra. a** Low-resolution survey spectrum of $C^2\Pi$ – $X^2\Sigma^+$ transition of $SiO^+$, before (lower) and after (upper) pumping toward $N = 0$. Two contaminating $Ba^+$ lines are present. The $v' = 3$ lines are expected to be broadened because of near-threshold predissociation, but these spectra are not yet fully understood. **b** Rotational fine structure of the $C^2\Pi_{1/2}$ – $X_2\Sigma^+$ transition (see Supplementary Information Section 1 for notation). **c** High-resolution spectra before and after pumping toward various $N$. The hot unpumped spectrum shows the sample is not yet thermalized to 300 K when pumping begins. **d** Fortrat diagram of the spectra, with marker areas proportional to deduced populations. Dashed and dotted lines are singly and doubly degenerate Fortrat parabolas. **e** Rotational populations corresponding to the spectra.

super rotors with narrow energy distributions offer new unique possibilities to study properties of these fascinating species over very long time periods and in isolation from unwanted environmental interactions. Here, the rotational state distributions were maintained by optical pumping for 10–20 min, limited only by chemical reaction with background $H_2$.

Molecular structure at high energies determines the long-range forces which play a crucial role in dissociation and reaction dynamics, for example determining the stability and lifetimes of reactive complexes. Diatomic potentials are commonly mapped by measurement of vibrational energies, but poor Franck-Condon overlap poses challenges for populating states near dissociation. However, centrifugal distortion parameters obtained from spectra of highly excited rotational states can equally well be used to map these regions[25]. Furthermore, direct measurement of dissociation rates provides information about the high-energy molecular potential. More generally, spectroscopically measured energies

and lifetimes of super rotors can be incorporated into direct potential fitting procedures[4] to recover more complex potential energy curves as well as non-adiabatic interactions of diatomic and polyatomic molecules.

Although here we exploited diagonal FCFs of $SiO^+$, broadband light sources are becoming more readily available and should facilitate extension of the optical pumping technique to molecules without such a transition, including polyatomics[26]. The only requirement for optical pumping is an excited state which decays to the state of interest faster than population is removed by collisions, blackbody radiation, or spontaneous emission. Collisions in a UHV ion trap environment are rare, blackbody radiation can be reduced if necessary by cooling the apparatus, and many molecules have target states with slow spontaneous emission because of the $\omega^3$ factor. Optical pumping of trapped molecular ions to pure rotational states could also be crucial for certain implementations of quantum information processing[27–30], some requiring pumping to

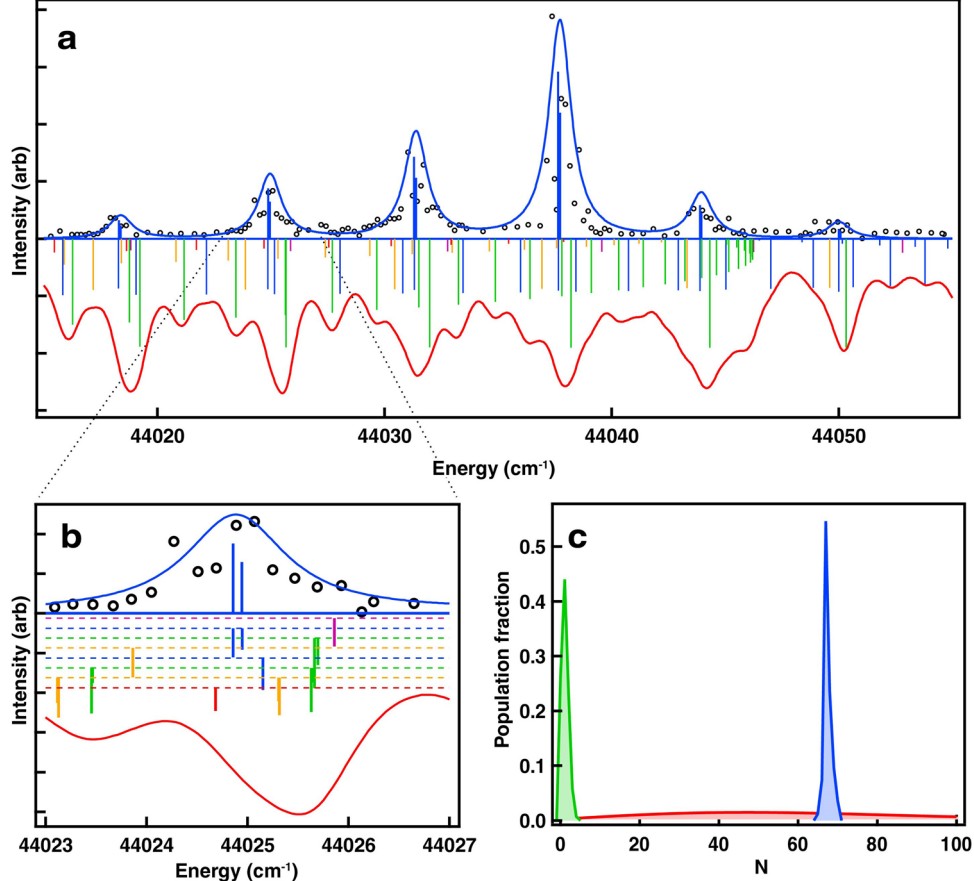

**Fig. 3 Optically pumped versus thermal spectra. a** Top trace — spectrum of SiO$^+$ recorded after pumping toward $N = 67$, showing resonances from r(65) through r(70); bottom trace — simulated spectrum of a thermal sample at 4600 K. Sticks show underlying rotational structure, color coded according to branch type (see Fig. 2b). **b** Close view of a region near the r(69) lines. Sticks in the thermal spectrum are vertically offset for clarity. **c** Rotational populations of the $N = 67$ ensemble (blue), $T = 4600$ K thermal sample (red) and $T = 3.6$ K thermal sample (green).

| Table 1 Measured spectroscopic constants of the C$^2\Pi$ state. | |
|---|---|
| **Parameter** | **Value (cm$^{-1}$)** |
| $T_e$ | 43644.4 (3) |
| $\omega_e$ | 982.5 (3) |
| $\omega_e x_e$ | 11.1 (3) |
| $B_e$ | 0.66925 (9) |
| $\alpha_e$ | 0.00722 (9) |
| $D_e$ | 1.24 (2)·10$^{-6}$ |
| $\beta_e$ | 4 (2)·10$^{-8}$ |
| $A_e$ | 179.8 (1) |
| $\alpha_A$ | −0.5 (1) |
| See Methods for details of the spectral fitting. | |

data points could be taken using only 10$^3$ – 10$^4$ molecules. Few-molecule spectroscopic sensitivity is critical for precision tests of physics beyond the Standard model[32] and parity violation[33,34]. Optical pumping combined with non-destructive state detection[35] and mass spectrometry could allow spectral identification at the single-molecule level and detection of abundances of the order of 10$^{-17}$ or less, many orders of magnitude beyond the best existing analytical methods. Such extreme sensitivity, currently available only for atomic trace isotope analysis[36], may find applications in many areas where ultrasensitive detection of molecules is needed. It would also allow detection of isomers of trace compounds (which cannot be separated based solely on M/z ratio) in the chemistry of combustion[37], the atmosphere[38], interplanetary missions[39], and forensics[40].

$N > 0$ in order to tune molecular splittings near resonance with trap frequencies[29]. In addition, optically pumped super rotors can be used for laboratory astrochemistry studies of their collisional and reactive properties, and their relaxation pathways in space[31].

Generally speaking, whether preparing ground or excited states of molecules, optical pumping aids in achieving the quantum projection limit of spectroscopy and more sensitive trace detection. In this work, all SiO$^+$ molecules were eventually pumped into the probed state and photodissociated, dramatically improving the signal-to-noise ratio. Without optical pumping, only a small fraction could have been probed at any specific dissociation wavelength. Thus, a spectrum of several hundred

## Methods

Unlike the optical centrifuge, which drives population to super rotors states coherently via stimulated Raman processes, optical pumping is incoherent and relies on many cycles of excitation followed by spontaneous emission. When pumping to super rotor states, the net result is increasing the rotation by many quanta. Super rotors are produced by optical pumping of the B,0-X,0 transition (see Fig. 6). Spectral coverage of R branch lines of results in adding angular momentum to the SiO$^+$ molecules. Spectral coverage of P branch lines removes angular momentum. Cut-offs are placed below the desired $N$ on the R branch and above the desired $N$ on the P branch, in order to create a dark state in the vicinity of $N$. Having cut offs on both sides drives population to the target $N$ and spontaneous emission removes entropy. When producing $N > 160$ super rotors, no cut off is placed on the P branch; instead we rely on fast radiative relaxation of rotational levels in the ground state, whose lifetime decreases as $1/\omega^3$ and is ~44 ms for $N = 160$.

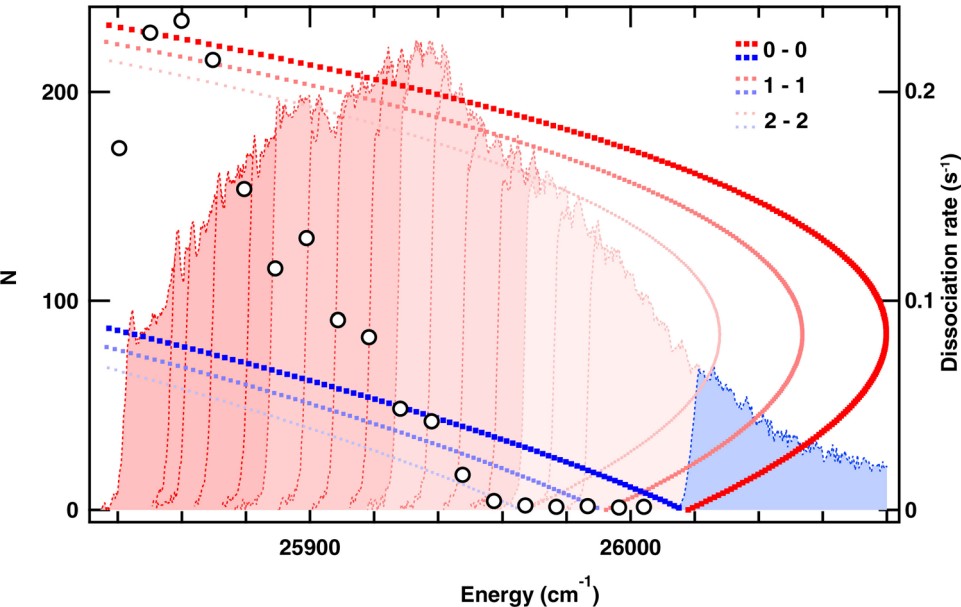

**Fig. 4 N > 160 two-step preparation and dissociation measurements.** The blue spectral area was exposed in Step 1, pink and red-shaded areas are added serially in Step 2. Blue and red dots, associated with the left-hand y-axis, are Fortrat parabolas of 0-0, 1-1 and 2-2 vibrational bands of the B–X transition. Open circles, associated with the right-hand y-axis, show the dissociation rates corresponding to the various Step 2 optical pumping spectra.

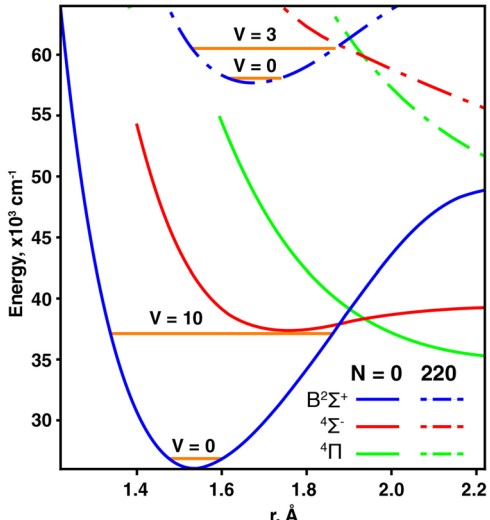

**Fig. 5 Using super rotors to probe molecular structure far from equilibrium.** MRCISD+Q/aug-cc-pwCV5Z potential energy curves (see Supplementary Information Section 5 for details) for $SiO^+$ at $N = 0$ and at $N = 220$, the latter altered significantly by centrifugal distortion. The $B^2\Sigma^+$ vibrational levels $v = 0$ and $v$ near the $^4\Sigma^-$ state crossing (at around $r = 1.9$ Å for both $N$) are shown.

$Ba^+$ and $SiO^+$ samples were loaded into the trap by means of ablation with the 532 nm Nd:YAG Minilite laser. Ba atoms were directly photoionized at 237 nm while SiO molecules were photoionized using a $1 + 1$ REMPI process via the 5–0 band of the A–X transition of SiO at 214 nm[41]. Photoionization UV light was generated with a tunable EKSPLA OPO system. $^{138}Ba^+$ ions were Doppler cooled by optical pumping via the $6^2S_{1/2}$–$6^2P_{3/2}$ transition at 493 nm and repumping the $5^2D_{3/2}$ "dark" state via the $5^2D_{3/2}$–$6^2P_{3/2}$ transition at 650 nm with continuous wave diode lasers. The translational temperature was sufficiently reduced so that the ions formed ordered structures known as Coulomb crystals. Since the effective trap potential is inversely proportional to the ion's mass, the Coulomb crystal that we observed on CCD camera had a dark core formed by $SiO^+$ ions and surrounded by bright $Ba^+$ ions.

$SiO^+$ reacts with the background gas comprised mainly of $H_2$ molecules, which are present in our UHV system at densities of $\sim 10^7$ cm$^{-3}$. The $SiOH^+$ molecules formed in the reaction stay trapped, and the LCFMS peaks of $SiOH^+$ and $SiO^+$ overlap. The $SiO^+$ lifetime in the trap is reaction-limited to 10-20 min. To avoid

build-up of $SiOH^+$, we dumped the reaction products and photofragments and loaded a fresh $SiO^+$ sample either every 5 min or when significant photo-dissociation occurred. Dumping was achieved by blue detuning the 493 nm $Ba^+$ cooling laser 5-25 MHz relative to the line center to "melt" the Coulomb crystal over several seconds until all light ions ($SiO^+$, $Si^+$, and $SiOH^+$) exit the trap. Due to much higher $M/z$ and lower secular frequency, $Ba^+$ ions were typically unaffected by the dumping routine and stayed in the trap.

We detected the ions loaded in the trap using an in-situ laser cooled fluorescence mass spectrometry (LCFMS) technique (Fig. S1). Ions in the quadrupole RF trap potential oscillate at a motional frequency inversely proportional to their mass. Motional oscillation of the ions in the trap was excited with a resonant low voltage (0.5–1 V) RF waveform applied to one of the trap rods. The excitation of ion's motion resulted in heating of the ion crystal and broadening of the $Ba^+$ 493 nm atomic line, leading to a depletion of the $Ba^+$ fluorescence. To achieve maximum fluorescence depletion, the 493 nm Doppler cooling laser was tuned 8-12 MHz to the red of the $Ba^+$ $6^2S_{1/2}$-$6^2P_{3/2}$ line center. We monitored the time-resolved $Ba^+$ fluorescence as a function of RF frequency, by photon counting with a Hamamatsu H8259 PMT. Typical depletion of $Ba^+$ fluorescence during excitation of $SiO^+$ secular frequency was 10–20%. The depletion was found to be proportional to the number of $SiO^+$ ions.

$SiO^+$ ions were photodissociated inside the trap by tunable UV light from 210 to 230 nm produced either by an Ekspla OPO (linewidth 4 cm$^{-1}$) or by a Scanmate 2 dye laser (linewidth 0.2 cm$^{-1}$). The OPO wavelength was calibrated using $Ba^+$ atomic lines, whereas the wavelength of the dye laser was measured using a Bristol 871 wavemeter. Photodissociation was detected using two different methods, both based on the depletion of $Ba^+$ fluorescence.

The "mass spectrometer method" was used for spectral surveys over wide wavelength ranges. In this method, an RF waveform with an amplitude of 1 V was swept over a range of 200–550 kHz for 200 ms, detecting both the $SiO^+$ and $Si^+$ resonances at 230 kHz and at 360 kHz, respectively. The photodissociation could be observed either by the depletion of the $SiO^+$ parent ion resonance or by the appearance of the $Si^+$ daughter ion resonance. Since $Si^+$ was found to leave the trap on a time scale of several seconds, the photodissociation signal was typically recorded by monitoring the parent ion channel. The integrated signal under the $SiO^+$ peak was averaged for 50 sweeps taken over 10 s. The photodissociation signal was obtained by subtracting these averages taken before and after photodissociation.

The "steady-state detection method" achieves higher signal-to-noise over narrow spectral ranges. The $SiO^+$ was photodissociated with a pulsed laser with a repetition rate of 10 Hz. The $SiO^+$ motional frequency was excited with a waveform consisting of 8 frequencies separated by 2.5 kHz and centered at 230 kHz. The excitation was applied for a period of 40 ms followed by a 60 ms period of "silence" for a background measurement. The photodissociation laser pulses arrived at the beginning of the motional excitation window. If photodissociation occurred, the $Ba^+$ fluorescence depletion exponentially decayed with time provided the laser fluence was set low enough. The decay constant is proportional to the product of line strength and fractional population in the probed energy level (see Supplementary Information Section 4).

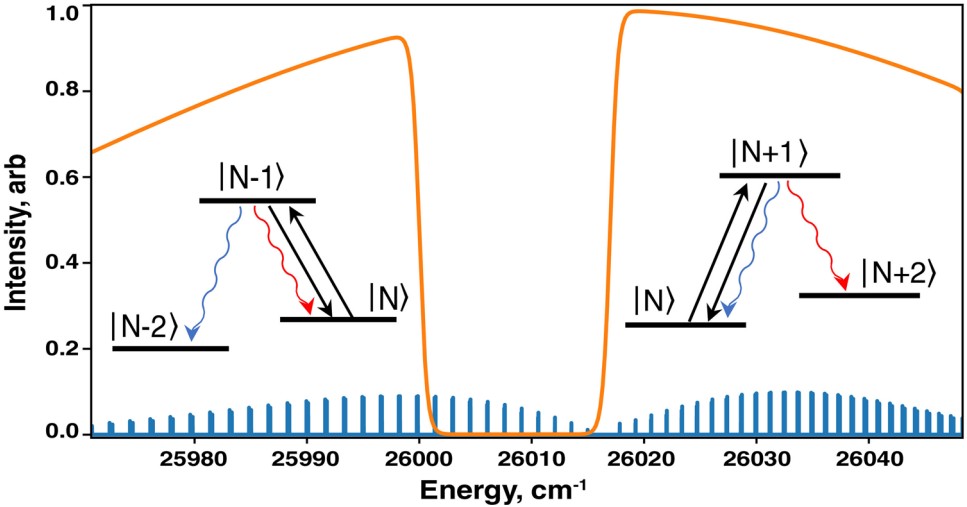

**Fig. 6 Optical pumping schematic.** Repeated excitations on successive $\Delta N = +1$ lines (R branch) in the right part of the spectrum accelerate SiO$^+$ rotation, while those on $\Delta N = -1$ (P branch) in the left part of the spectrum decelerate SiO$^+$ rotation.

We fitted to Lorentzian line shapes because our translational temperature was very low due to sympathetic cooling (<1 K), and lifetime broadening due to fast dissociation of the C state dominated line profiles. Doppler broadening at 1 K for the SiO$^+$ 44,000 cm$^{-1}$ transition is ~100 MHz, and lifetime broadening even for the most stable C, v = 0 level is of order of 1.5 GHz. We used PGOPHER for the fit, which had expressions for line energies and intensities of a $^2\Pi$ - $^2\Sigma$ transition, and rotational populations were among the fit parameters along with molecular constants.

## Data availability
All data is available upon request.

## Code availability
All code is available upon request.

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

## Acknowledgements

We gratefully acknowledge enlightening conversations with Michael Heaven and Valery Milner.

## Author contributions

I.A. and P.S. contributed equally to the work. I.A., P.S., and S.V. developed the laboratory techniques and took data. B.O. led the experimental effort. A.P.L.B and A.G.S.O.-F. performed the theoretical calculations.

## Funding

Development of dissociative analysis techniques were funded by NSF Grant No. PHY-1806861, and development of state control for super rotor states was funded by ONR, Grant No. N00014-17-1-2258. A.G.S.O.-F acknowledges São Paulo Research Foundation (FAPESP, 2020/08553-2) and the Conselho Nacional de Desenvolvimento Científico e Tecnológico (CNPq) of Brazil (306830/2018-3). A.P.L.B and A.G.S.O.-F thank the Coordenação de Aperfeiçoamento de Pessoal de Nível Superior - Brasil (CAPES) - Finance Code 001.

## Competing interests

The authors declare no competing interests.
