## [Peer Review File · Nature Communications]

Reviewers' Comments:

Reviewer #3:

Remarks to the Author:

The work describes the generation of highly rotationally excited SiO⁺ molecules (super rotors) using optical pumping with a broadband laser. The molecules were confined in a linear Paul trap and cooled in a Coulomb crystal via interaction with laser cooled Ba⁺ atoms. As a result, a narrow distribution of super rotors was formed centered around $N = 67$; the work also produced earlier unknown molecular constants for the C state of SiO⁺. The experiments are exciting and neat; the work benefits from the development of the greatly enhanced spectroscopic sensitivity provided by rotational state control in a collision-free environment, which allows to record spectra only from as few as $10^3 - 10^4$ molecules. I am impressed with these experimental achievements and interesting findings of this paper and the calculations of the pertinent potential energy surfaces are carried out using state-of-the-art levels of theory. However, to justify publication in Nature Communications, the authors should make a better case why this work is fundamentally significant and why it is interesting to a broad scientific readership. The statements on this matter made in abstract: "Improved optical control of molecular quantum states promises new applications including chemistry in the quantum regime, precision tests of fundamental physics, and quantum information processing" and Conclusions: "... extreme sensitivity, currently available only for atomic trace isotope analysis, may find applications in many areas where ultrasensitive detection of molecules is needed, e.g. studying transient reaction intermediates in combustion, atmosphere, detection of trace species in interplanetary missions, and forensic chemistry" do not seem substantiated by the content of the work. A reader must see a clearer connection how the generation of SiO⁺ super rotors by optical pumping will help e.g., developing quantum information processing or detection of transient combustion intermediates or trace species in interplanetary missions (which usually are not simple diatomic molecules). Only if this case is clearly made, I would recommend publication of this article in Nature Communications.

Reviewer #4:

Remarks to the Author:

The manuscript titled, "Precisely Spun Super Rotors" describes a technique for producing a very narrow distribution of highly rotationally excited SiO⁺ molecules in excited electronic states. This work has two novel contributions to the literature. First, few super rotor studies have used optical traps in order to translationally cool the molecules before rotational excitation. Second, there has been very little work examining super rotor molecules in excited electronic states. This offers the prospect for testing beyond the Born-Oppenheimer approximation. This paper is well presented and deserves publication. However, there are a few comments that should be addressed before publication.

1. It is not clear to the reader exactly how the molecules are being rotationally excited. The authors state that they use broadband tailored spectra in order to cool and excite molecules to a specified state. How can you excite molecules to such a narrow distribution of high rotational states given the rotational selection rules? How come the molecules become concentrated around a single state? What is the process by which the ensemble of molecules become narrowly distributed in a particular rotational state?
2. Fig. 1 D shows the spectra used to excite the molecules. However, the legend at the right is confusing. It is unclear what is meant by the various shades of color and to which state the molecules are being driven.
3. How does the value of the centrifugal distortion in the C2PI state compare to the ground and B electronic states? What does that say about the molecular potential in the C2PI state?
4. Fig 4. Is a little confusing. What is the dissociation rate plot telling us? There is a lot going on in this plot. Why are there several dots near $N=0$ with different energies?
5. In Fig. 5 how are these potential energy curves calculated? This is explained in the supplemental information, but it would be helpful to have some idea where these curves are coming from in the main text. Why is the 4PI state being plotted? There is no mention of this state in the main text.

6. How are the rotational populations calculated? Do you fit the collected spectra to Gaussian functions in order to determine the translational temperature?

We thank the reviewers for their helpful comments. Our responses are written, in black text, below.

Reviewer #3 (Remarks to the Author):

The work describes the generation of highly rotationally excited SiO⁺ molecules (super rotors) using optical pumping with a broadband laser. The molecules were confined in a linear Paul trap and cooled in a Coulomb crystal via interaction with laser cooled Ba⁺ atoms. As a result, a narrow distribution of super rotors was formed centered around $N = 67$; the work also produced earlier unknown molecular constants for the C state of SiO⁺. The experiments are exciting and neat; the work benefits from the development of the greatly enhanced spectroscopic sensitivity provided by rotational state control in a collision-free environment, which allows to record spectra only from as few as $10^3 - 10^4$ molecules. I am impressed with these experimental achievements and interesting findings of this paper and the calculations of the pertinent potential energy surfaces are carried out using state-of-the-art levels of theory. However, to justify publication in Nature Communications, the authors should make a better case why this work is fundamentally significant and why it is interesting to a broad scientific readership. The statements on this matter made in abstract: "Improved optical control of molecular quantum states promises new applications including chemistry in the quantum regime, precision tests of fundamental physics, and quantum information processing" and Conclusions: "... extreme sensitivity, currently available only for atomic trace isotope analysis, may find applications in many areas where ultrasensitive detection of molecules is needed, e.g. studying transient reaction intermediates in combustion, atmosphere, detection of trace species in interplanetary missions, and forensic chemistry" do not seem substantiated by the content of the work. A reader must see a clearer connection how the generation of SiO⁺ super rotors by optical pumping will help e.g., developing quantum information processing or detection of transient combustion intermediates or trace species in interplanetary missions (which usually are not simple diatomic molecules). Only if this case is clearly made, I would recommend publication of this article in Nature Communications.

We have entirely reworked the Conclusions section to address the questions of this reviewer. This reworked section should be read in its entirety, but for convenience, below are some of the relevant points:

- We discuss the feasibility of optical pumping of more general species. The case that this can be done for many molecules is straightforward to make.
- It is of interest to extend this work and create super rotors of different species, not only for understanding their long-range potential energy surfaces as we have done here, but also to understand their relaxation pathways. The latter problem is relevant to astrochemistry, and we have included a new reference on this point.
- On the question of relevance to quantum information processing with trapped molecules, we now explicitly point out that in some proposals optical pumping to excited states (rather than the ground state) is a key feature.

- We must also return to the broader narrative—scientists are learning how to gain quantum control over molecules, thereby improving spectroscopy and detection. For some applications we discuss, pumping of molecules to their ground state would be just as good as pumping to excited states. In that case, the broadband techniques we develop here are still very relevant. But in the context of the general trend in improved quantum control, that optical pumping could be used to obtain super rotor states is non-obvious (as one question of Reviewer #4 highlights). This part of the story is well worth telling to a broad audience.

Reviewer #4 (Remarks to the Author):

The manuscript titled, “Precisely Spun Super Rotors” describes a technique for producing a very narrow distribution of highly rotationally excited SiO⁺ molecules in excited electronic states. This work has two novel contributions to the literature. First, few super rotor studies have used optical traps in order to translationally cool the molecules before rotational excitation. Second, there has been very little work examining super rotor molecules in excited electronic states. This offers the prospect for testing beyond the Born-Oppenheimer approximation. This paper is well presented and deserves publication. However, there are a few comments that should be addressed before publication.

1. It is not clear to the reader exactly how the molecules are being rotationally excited. The authors state that they use broadband tailored spectra in order to cool and excite molecules to a specified state. How can you excite molecules to such a narrow distribution of high rotational states given the rotational selection rules? How come the molecules become concentrated around a single state? What is the process by which the ensemble of molecules become narrowly distributed in a particular rotational state?

This is an excellent question which we did not make clear enough. It is also a good point to raise about the importance of this work. We rewrote the last paragraph of the Introduction so that it now includes the following. “Taking into consideration angular momentum selection rules, it is not at first obvious that optical pumping can be used to create super rotor states. However, this goal can be achieved by driving many repeated cycles of absorption and spontaneous emission on subsequent spectral lines. To this end, we use a broadband spectrum which is filtered⁽¹⁵⁾ such that no absorption occurs from the target rotational state N, but other states have their P or R branch transitions illuminated such that population is driven toward the target state.”

2. Fig. 1 D shows the spectra used to excite the molecules. However, the legend at the right is confusing. It is unclear what is meant by the various shades of color and to which state the molecules are being driven.

The figure caption now includes the following text. “(D) Spectrum of the 385 nm light used for pumping SiO⁺ toward N=0 (top) and N>0 (bottom). Blue (red) sticks represent transitions in the P (R) branch. Arrows to the right show the flow of optical pumping, with blue (red) representing lowering (raising) N, and shading represents intensity of the light.”

3. How does the value of the centrifugal distortion in the C2PI state compare to the ground and B electronic states? What does that say about the molecular potential in the C2PI state?

We added the following text. “The value of the centrifugal distortion of the C state is slightly higher than those of X and B states, which is consistent with its longer equilibrium bond length. Its agreement with Kratzer relationship(25) signifies that the C state potential near equilibrium is well described by a Morse potential.”

4. Fig 4. Is a little confusing. What is the dissociation rate plot telling us? There is a lot going on in this plot. Why are there several dots near N=0 with different energies?

The last two sentences of the figure caption were changed to the following. “Blue and red dots, associated with the left-hand y-axis, are Fortrat parabolas of 0-0, 1-1 and 2-2 vibrational bands of the B-X transition. Open circles, associated with the right-hand y-axis, show dissociation rate corresponding to the various optical pumping spectra.”

5. In Fig. 5 how are these potential energy curves calculated? This is explained in the supplemental information, but it would be helpful to have some idea where these curves are coming from in the main text. Why is the 4Π state being plotted? There is no mention of this state in the main text.

The figure caption now reads: “MRCISD+Q/aug-cc-pwCV5Z potential energy curves (see SM for details) for SiO+ at N=0 and at N=220, the latter altered significantly by centrifugal distortion. The B $2\Sigma^+$ vibrational levels v=0 and v near the 4 Σ^- state crossing (at around r = 1.9 Å for both N) are shown.”

And the following has been added to the main text. “Although for this range of N the molecules are bound in the X $2\Sigma^+$ state, dissociation from B $2\Sigma^+$ can occur through coupling to the ground state asymptote via interactions with excited electronic states, such as 1 $4\Sigma^-$ or 1 4Π (Fig. 5).”

6. How are the rotational populations calculated? Do you fit the collected spectra to Gaussian functions in order to determine the translational temperature?

A reference to Methods has been added to the main text, and the Methods now includes the following. “We fitted to Lorentzian line shapes because our translational temperature was very low due to sympathetic cooling (< 1 K), and lifetime broadening due to fast dissociation of the C state dominated line profiles. Doppler broadening at 1 K for the SiO+ 44000 cm $^{-1}$ transition is ~100 MHz, and lifetime broadening even for the most stable C, v=0 level is of order of 1.5 GHz. We used PGOPHER for the fit, which had expressions for line energies and intensities of a $2\Pi - 2\Sigma$ transition, and rotational populations were among the fit parameters along with molecular constants.”

Reviewers' Comments:

Reviewer #3:

Remarks to the Author:

The authors have properly addressed the comments of all reviewers and I recommend publication of this paper in Nature Communications in its present form.

Reviewer #4:

Remarks to the Author:

The authors have addressed most of my concerns. However, I still feel that more details could be provided about the rotational excitation process. The authors did add details in the introduction, but perhaps more details can be given in the Methods section. Alternatively, a figure showing the excitation process with an energy level diagram could be helpful. Other than that, the paper is well-presented and has very interesting results and so can be published as is.

Matthew Murray